# The Potential Global Distribution of the White Peach Scale *Pseudaulacaspis pentagona* (Targioni Tozzetti) under Climate Change

**Yunyun Lu [1], Qing Zhao [1], Lifang Cheng [1], Ling Zhao [1], Hufang Zhang [2] and Jiufeng Wei [1,\*]**

[1] Department of Entomology, Shanxi Agricultural University, Taigu 030801, China; 17835424016@163.com (Y.L.); Zhaoqing86623@163.com (Q.Z.); LFC3302@163.com (L.C.); 18734422759@163.com (L.Z.)

[2] Department of Biology, Xinzhou Teachers University, Xinzhou 034000, China; Zh_hufang@sohu.com

[\*] Correspondence: wjfeng@nwsuaf.edu.cn

**Abstract:** The white peach scale *Pseudaulacaspis pentagona* (Hemiptera: Diaspididae) is a pest that causes significant damage to more than 221 genera of host plants in more than 112 countries. *P. pentagona* primarily feeds on mulberry, peach, and tea, and this leads to the loosening of the epidermis of trees, which damages nutrient and water transportation in the branches, leading to branch death. *P. pentagona* is native to China and Japan, and has become an invasive species all over the world. However, the potential distribution of *P. pentagona* remains unclear. In this study, a potential distribution map of *P. pentagona* was developed using current and future climate information using MaxEnt. The model indicates that Asia, Europe, South America and North America are a highly suitable habitat range for this species. The MaxEnt models for the potential distribution of *P. pentagona* for the 2050s and 2070s suggest that in the case of no significant increase or even decrease in the highly suitable area, the suitable area increased significantly on any future climatic scenarios. The predicted area gain in the suitable habitat is $2.82 \times 10^7$ km$^2$, including more of Asia, such as China, Japan, and Mongolia, and also including India, Vietnam, Romania, Ukraine, Poland, Hungary, Austria, The Czech Republic, Italy, and Germany in Europe, which shows an increase of 24.5% over the current habitat on RCP8.5 emission scenarios for the 2070s. With the warming of the climate, significant expansions are predicted in the suitable area, especially in Europe and East Asia. Under RCP8.5 for the 2050s, the model-predicted that the area of suitable habitat in China and the Korean Peninsula gains an increase of 18.8% over the current suitable habitat area. Under other climate scenarios, RCP8.5-2070s, the suitable areas were the largest, compared to projection for the current climate scenario (ca. 24.1% increase) which increased to $7.89 \times 10^6$ km$^2$. In Europe, under RCP8.5 for the 2070s, the highly suitable areas were the largest, compared to the projection for the current climate scenario (ca. 46.2% increase), which increased to $8.64 \times 10^5$ km$^2$, the area of suitable habitat suitability increased to $4.99 \times 10^6$ km$^2$ (29.2% increase of the current condition). Potential increases or decreases in distribution ranges were modeled under future climatic scenarios. This study suggests that the most important factor that influenced current distribution of this pest was temperature, and BIO3 (isothermality) was the most important factor that contributed to 48.6% of the potential distribution map. Given the rapid spread of *P. pentagona* and the serious risk this species poses to local ecosystems, warning modelling and practical strategies to prevent the establishment and expansion of this species should be sought. This distribution map will help governments to identify areas that are suitable for current and future infestations, and to optimize pest management strategies.

**Keywords:** pest; white peach scale; species distribution model; risk assessment; habitat suitability; pest management

## 1. Introduction

Insects are the most widely distributed animals in the world, constituting approximately half of the global biodiversity [1]. Invasive insect species have spread around the world through the expansion of the trade, tourism and material exchange, which has caused significant damage [2]. With the increasing rate of global trade and human activity, the probability of invasion by non-native pest species is on the rise [3]. Invasive species cause economic damage by damaging productivity of agriculture and human health, and also by threatening global biodiversity, as they can cause the extinction of endemic species [4]. The control and management of pest insects cost a minimum of US $70.0 billion per year, and the biggest agricultural producers, such as China and the United States, experience the greatest absolute cost for managing invasive species [5].

Scale insects are major agricultural and forest pests in the United States, as they do not have natural enemies, causing serious ecological damage [6]. The white peach scale *Pseudaulacaspis pentagona* (Targioni-Tozzetti) (Hemiptera: Diaspididae), is a common pest in orchards. *P. pentagona* is an extremely invasive and can feed on 221 genera of host plants distributed among 85 plants families in more than 112 countries [7]. *P. pentagona* mainly harms plants belonging to the Rosaceae family [8], such as mulberry, and has caused significant damage to the silk industry [9,10]. In addition, *P. pentagona* also harms to fruit trees [11], ornamental plants, such as common box *Buxus sempervirens*, cherry laurel *Prunus laurocerasus*, *P. amygdalus*, golden currant *Ribes aureum*, false acacia *Robinia pseudoacacia*, *Tamarix* sp and Japanese pagota tree *Sophora japonica* [10,12] and wild plants, such as *Zamia* spp., *Ulmus* spp., tree of heaven *Ailanthus altissima* and *Spondias* spp. [10,13,14]. *P. pentagona* can cause production losses of up to $480,000 in untreated pear orchards in the United States [15], and papaya plants severely infected with *P. pentagona* in Hawaii have caused serious quarantine problems [16]. The nymphs of *P. pentagona* usually settle and feed on the roots of forest plants. Successful generations form dense aggregates that gradually spread upwards on the stem and foliage. Their feeding weakens the plant and may cause cracking of the bark, which may subsequently lead to the destruction of host and forest [17]. With increasing trade, the distribution of *P. pentagona* has increased around the world. As a result, this species is considered a quarantine pest in many countries, including Bulgaria [18], Fiji [19], and the United States [6]. Scale insects damage plants by piercing the branches and sucking on the sap, which damages the areas that the insects used for feeding. These insects often feed in clusters, forming white flocs around the branches, and this leads to the loosening of the epidermis of the trees, affecting the transportation of nutrients and water, and causing in branch death [12].

*P. pentagona* is native to Asia, although it was first discovered and described in Italy by Tranfaglia in 1886 [20]. It is now widely distributed in 112 countries in temperate, subtropical and tropical regions, including Europe, Australia, Africa [7]. In 1902, Gossard first reported the damages caused by *P. pentagona* Florida, USA. This species subsequently spread to Texas in the west, and Tennessee and Maryland in the north [21].

The last assessment report from the Intergovernmental Panel on Climate Change (IPCC) has predicted an increase in mean global temperature from 1.1 to 5.4 °C by the year 2100 [22]. Increase in global temperature will cause impacts agricultural production around the world [23]. Recent studies have demonstrated that climate warming may affect the life history and population dynamics of insects, altering traits such as developmental rate, chemistry and distribution range [24,25]. Increased temperature can affect the physiological characteristics of insects, thereby affecting the distribution of pests through changes in their suitable range [26]. Pests are extremely sensitive to climatic conditions, and global climate change may potentially exacerbate the outbreak of pests [27].

While *P. pentagona* is found in more than 112 countries, its specific distribution range remains unclear. The distribution range of a pest directly affects management strategy, therefore it is important to understand the potential distribution of a pest. Environmental (or ecological) niche models provide an effective way identify distribution patterns of insects [28]. A variety of models have been derived to predict the distribution of species, such as GLM (Generalized Linear Model), GAM (Generalized Additive Model) [29,30], GARP (Ganeral Rule Set Pruduction) [29,31], MaxEnt (Maximun entropy) [32],

and Bioclim (Bioclimatic envelope) and DOMAIN [29]. MaxEnt generally provides stable operation results, short calculation time, and can be conducted using reasonable computational resources, while fitting complex models using smaller datasets. MaxEnt achieves this by using implicit "regularization" mechanisms to prevent model complexity from increasing beyond what is supported by empirical data, thereby stimulating distribution of species under more realistic habitat conditions [32].

Identifying potential distribution areas of pests under different climate scenarios is necessary to develop strategies to control and limit the expansion of pests [33]. Environmental niche models provide a powerful tool to manage pests. Risk map outlines the suitability of new species invasion areas, and can be used to prevent or control invasion. However, methods used to develop risk maps are variable and have not been standardized [34]. Previous studies have focused on the biological characteristics and damages caused by *P. pentagona*, and currently we do not know the predicted distribution area for this species [13,35]. Thus, we used the known species occurrence data and different climate change scenarios to predict current and future potential distribution areas of *P. pentagona* by addressing two goals: (1) Generating risk maps based on current and future distribution under different climate change scenarios, and (2) Identifying the factors that create suitable habitats for *P. pentagona*.

## 2. Materials and Methods

### 2.1. Species Occurrence Data

Geographical occurrence data of *P. pentagona* were obtained from the following resources: the Center of Agriculture and Bioscience International (CABI, https://www.cabi.org), Global Biodiversity Information Facility (GBIF, https://www.gbif.org/), European and Mediterranean Plant Protection Organization (EPPO, https://www.eppo.int), and Scalenet (http://scalenet.info/), as well as information in the literature and field collection data. Geo-coordinates for each data point were obtained based on the specific location of a collection site (accurate to county) and using Google Maps (https://www.google.com/maps/) to identify coordinates. Data points with unclear or incorrect information were deleted, yielding 415 occurrence records (Table S1). The literature used to identify the occurrence data were listed in Table S2.

Collection sites are often biased towards areas that are easy to access, such as near a city. This causes the occurrence data to have different sampling intensity. We used an established technique called fishnet to divide the data into grids with an area of −25 km$^2$. We randomly select one record per grid if the grid included more than one record. After filtering, the occurrence data was reduced to 349 records [36]. The list of species occurrence and the distribution map are listed Table S3. The organization of data was achieved using ArcGis 10.1 (ERSI 2012, RedLands, CA, USA) (http://www.esri.com/).

### 2.2. Environmental Variables

2.2.1. Current Environmental Variables

Climate change can affect the persistence of insect species, especially those with a narrow habitat, and can force them to either adapt to the new conditions or shift their geographical distribution [37]. There are 19 commonly used layers when modeling the effects of climate on species distribution, referred to as the "Bioclim" layers, and they include temperature, precipitation, and seasonality. These variables (Bio1-Bio19) (www.worldclim.org/ 2.5 min) were used to determine environmental parameters that may have the greatest effects on the distribution of *P. pentagona*. In all predictive modeling, it is prudent to select explanatory variables that are not closely correlated [38]. We used correlation analysis to select environmental variables for our species distribution modelings (SDMs) [39]. The purpose of selecting layers is, that many previous researches have shown that highly correlated variables affect the accuracy of predicted results for species distribution modellings (SDMs). Annual precipitation has long been recognized as a major determinant of species distributions [40]. Temperature determines the spread and suitability of habitats, and also induces physiological changes, such as dormancy,

diapause, aestivation and hibernation [41,42]. The multicollinearity of ecological variables will affect the prediction results, resulting in over fitting of model results. In order to remove the influence of multicollinearity on the model results, we use Pearson correlation coefficient method to analyze the correlation of the 19 environmental variables based on SPSS Statistics 17. We included only variables with a Pearson correlation coefficient of less than 0.8 (|r| < 0.8) for each pairwise comparison of all 19 climatic variables and removed highly correlated variables. Finally, the variables selected for predicted modelling were BIO2 (Mean Diurnal Range), BIO3 (Isothermality), BIO8 (Mean Temperature of Wettest Quarter) and BIO15 (Precipitation Seasonality) (Table S4).

### 2.2.2. Future Environmental Variables

For our future climatic predictions, to account reduce uncertainty in the species distribution future scenarios, three random Global Circulation Models (GCMs) were obtained from the Worldclim database with a spatial resolution of 2.5 arc min. These were randomly selected to account for the uncertainty of future climate scenarios [43]. The Hadley Global Environment Model 2-Atmosphere Ocean (HADGEM2-AO), Beijing Climate Center Climate System Model (BCC-CSM1-1) and the Model for Interdisciplinary Research on Climate (MIROC5) for 2050 (average for 2041–2060) and 2070 (average for 2061–2080) including representative concentration pathways (RCPs; RCP2.6, RCP4.5, RCP6.0 and RCP8.5) were obtained from the Coupled Model Inter-comparison Project Phase 5 (CMIP5) from International Panel on Climate Change (IPCC) [44]. A final suitability map was created by averaging the maps from the three future climate scenarios to reduce the uncertainty among GCMs [45]. We selected two mean periods (2050s and 2070s) to predict the future distribution of *P. pentagona*.

### 2.3. Environmental Niche Models

Species distribution models combine information from species occurrence data and environmental variables to predict the geographical distribution of a species. Various statistical approaches have been employed in this type of modeling [46].

ENMs (Ecological niche modelings) were generated using the maximum entropy algorithm using MaxEnt version 3.3.3. We used the maximum entropy method (Maxent) to model geographic distributions of *P. pentagona* using presence-only data. Maxent is a general-purpose machine learning method with a simple and precise mathematical formulation, and it has a number of aspects that make it well-suited for modeling species distribution [31]. We selected this software for the following reasons: (1) MaxEnt Models are widely used in species distribution predictions [47]; (2) it is better at predicting the effect of climate change on species distribution and suitable area division compared to other ENMs, such as GARP and BLOCLIM [48]; (3) Maxent model results are more conservative [49]; and (4) Maxent estimates the distribution (geographic range) of a species by finding the distribution that has maximum entropy (i.e., is closest to geographically uniform), and this can be constrained based on the environmental conditions at recorded occurrence locations [50,51].

The MaxEnt model applies a machine learning method called maximum entropy modeling, which follows the principle of maximum entropy. We show the formula for maximum entropy as follows:

$$H(\hat{\pi}) = -\sum_{x \in X} \hat{\pi}(x) ln\hat{\pi}(x)$$

The unknown probability distribution, which we denote $\pi$; the approximation of $\pi$ is $\hat{\pi}$; $X$ is a finite set; $x$ is the individual elements of $X$ as points; and ln is the natural logarithm. Typically, feature types and regularized multipliers are used to optimize the model and control over-parameterization using MaxEnt. These feature types represent different transformations of covariates, including linear, product, hinge, threshold, and quadratic features, which allows the software to optimize for the species of interest and prevent overly simple or complex models. In order to produce the best model, we used the R package "ENMeval" to test whether parameters were over-fitted, and selected multipliers and combinations of feature classes based on these results [49]. After partitioning occurrence data using

the checkerboard2 method with aggregation factor of 5, we built models with RM values ranging from 0.5 to 4.0 in increments of 0.5, and with six different FC combinations (L, LQ, H, LQH, LQHP, LQHPT; where L = linear, Q = quadratic, H = hinge, P = product and T = threshold) [52]. We used the "checkerboard2" approach to calculate the Akaike information criterion coefficient (AICc), and the lowest delta AICc scores were selected to run the final MaxEnt models. The results are shown in Figure S1 and Table S5. The regularization multiplier was set to 1.5 and the QHPT feature combinations was selected for our analysis. The logistic output of ENMeval was used to run the MaxEnt model. The 10th percentile replicate training presence logistic threshold was used to define the suitable and unsuitable habitats for *P. pentagona*. This threshold is widely used in species distribution modeling, especially when the observation datasets were collected by different observers and methods over a long period of time [53]. The potential distribution map area of *P. pentagona* was divided into four levels using the above threshold (0.2168): < threshold denotes unsuitable habitat; threshold—0.4 denotes low habitat suitability; 0.4–0.6 denotes moderate habitat suitability; and 0.6–1 denotes high habitat suitability. Finally, 10-fold replicate cross-validation was used to run MaxEnt to prevent random errors from the predicted samples.

*2.4. Model Evaluation*

The area under the ROC (Receiver Operating Characteristics) curve, or simply AUC, has been used in medical studies since the 1970s to evaluate prediction models. AUC is a better measure for establishing objective criteria and currently used to assess the accuracy of predictive distribution models. This suggests that AUC should replace accuracy in measuring and comparing classifiers [54,55], as it avoids the supposed subjectivity in the threshold selection process. However, AUC has some disadvantages for our approach, as it ignores the predicted probability values and the goodness-of-fit of the model, and it weights omission and commission errors equally. Therefore, we chose a partial receiver operating characteristic (pROC) metric approach to assess the robustness of the model using Niche Toolbox (http://shiny.conabio.gob.mx:3838/nichetoolb2/) with 1000 replicates and E = 0.05. Finally, we displayed the data using ArcGis.

## 3. Results

*3.1. Statistical Model Performance*

Based on known occurrences of *P. pentagona* and current climate layers, we predicted the current suitable distribution map of *P. pentagona*. The model performance test yielded optimal results for partial ROC (mean value AUC: 0.9526019) and the distribution of AUC ratios calculated as AUCpartial/AUCrandom was significantly higher than random expectations, therefore our model showed high performance ($p < 0.0001$) (Figure S2). A 10th percentile training presence logistic threshold value of 0.2168 was obtained for the map predicting current distribution of *P. pentagona*. In this map, threshold below 0.2168 indicates unsuitable habitat for this species.

*3.2. Current Potential Distribution*

Based on the current species occurrence data and climate variables, we determined the current risk potential distribution maps for *P. pentagona* (Figure 1).

The current distribution map suggested that central North America, eastern South America, northern Africa, central Europe, and most east southern and central parts of Asia have suitable environmental conditions for *P. pentagona.* In North America, central, Northeastern, North Atlantic and Gulf Coast are key distribution points. In South America, the eastern coast of Brazil was more suitable for *P. pentagona.* In Europe, highly suitable areas for *P. pentagona* were Hungary, eastern Croatia, Slovenia and Slovakia, northern and southern Austria, southwestern Czech Republic, northern Italy, southwestern Romania, southern Moldova, northern and central Bulgaria and Serbia, the junction of Germany and France, central France, Barcelona region and east coast of Spain, Black Sea coast of

southern Ukraine, the junction of Russia Georgia junction, as well as central Georgia and Azerbaijan. In Africa, the Eastern Mediterranean coast of Tunisia has a very high risk of being invaded by *P. pentagona*. In Asia, there were many areas that the model predicted as having high risk of being invaded by *P. pentagona*. This included northeastern Pakistan, northern India, Huabei regions of south China, Huazhong area of East China, Shaanxi Qinling area, the northern coast of Taiwan and Hainan of China, central and northern Vietnam and its adjacent areas in China, most of South Korea, Japan's Kyushu Island, south and central areas of the Shikoku Island, central and southern Honshu Island. Under current climate change scenarios, China, Japan, South Korea, North America and central and southern Europe have large regions that are predicted to have highly suitable habitats for *P. pentagona*. Based on current climatic variables, the total area of potential suitable habitat for *P. pentagona* is approximately $-2.26 \times 10^7$ km$^2$, of which $-3.83 \times 10^6$ km$^2$ (i.e., 16.9% of the total potentially invadable area) has high habitat suitability (high risk), and $-6.28 \times 10^6$ km$^2$ (approximately 27.8% of the total potentially invadable area) has moderate habitat suitability (Figure 2A).

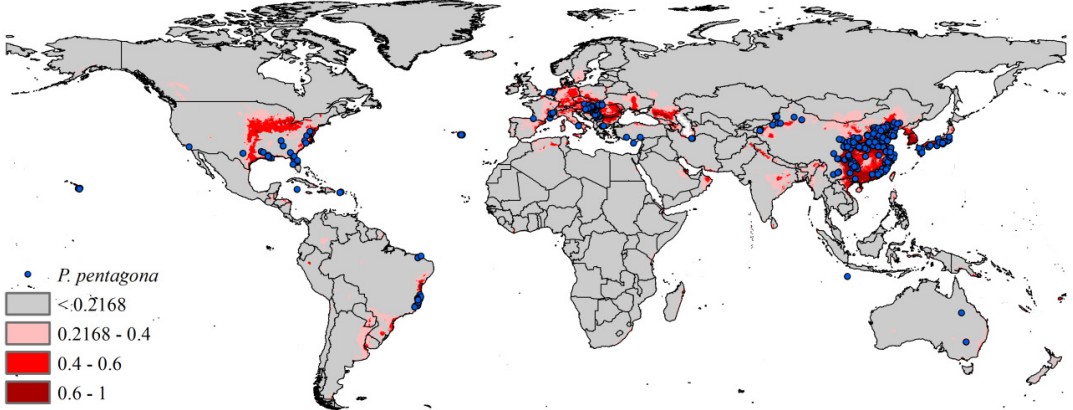

**Figure 1.** The occurrence sites and the potential distribution of *Pseudaulacaspis pentagona* used in current modeling. Blue point, occurrence sites; Gray, unsuitable habitat area; Pale red, low habitat suitability area; Red, moderate habitat suitability area; Dark red, high habitat suitability area.

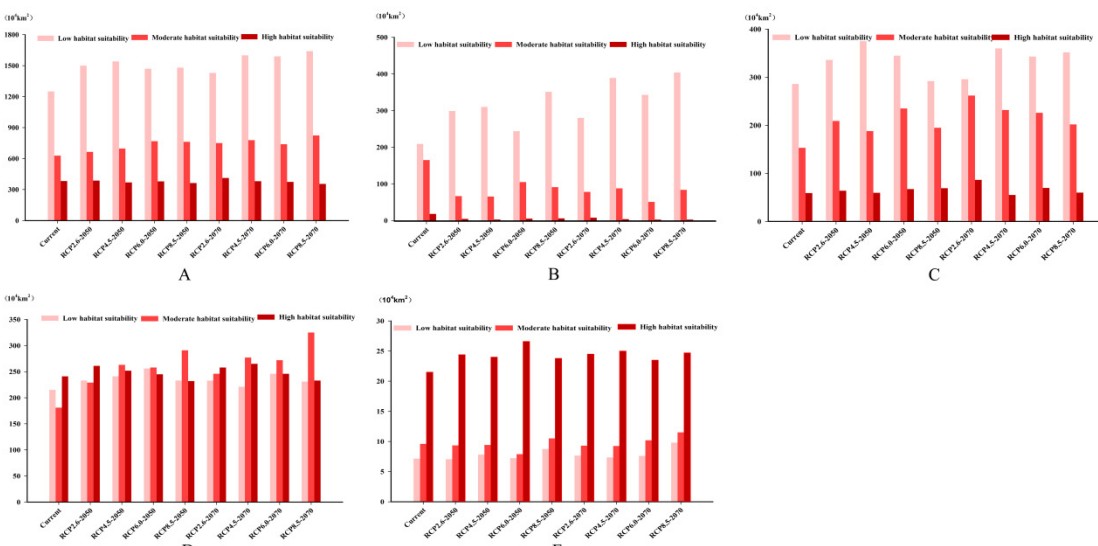

**Figure 2.** Predicted suitable areas for *Pseudaulacaspis pentagona* under current and future conditions (km$^2$) (**A**): Globe; (**B**): North America; (**C**): Europe; (**D**): China and Korean Peninsula; (**E**): Japan Pale red: Low habitat suitability; Red: Moderate habitat suitability; Dark red: high habitat suitability.

Based on the response curves (Figure 3), four environment variables that associated with highly habitat suitability area were 2–11.2 °C for BIO2, 2.6–3.9 °C for BIO3, 19–28 °C for BIO8 and 15–260 mm for BIO15.

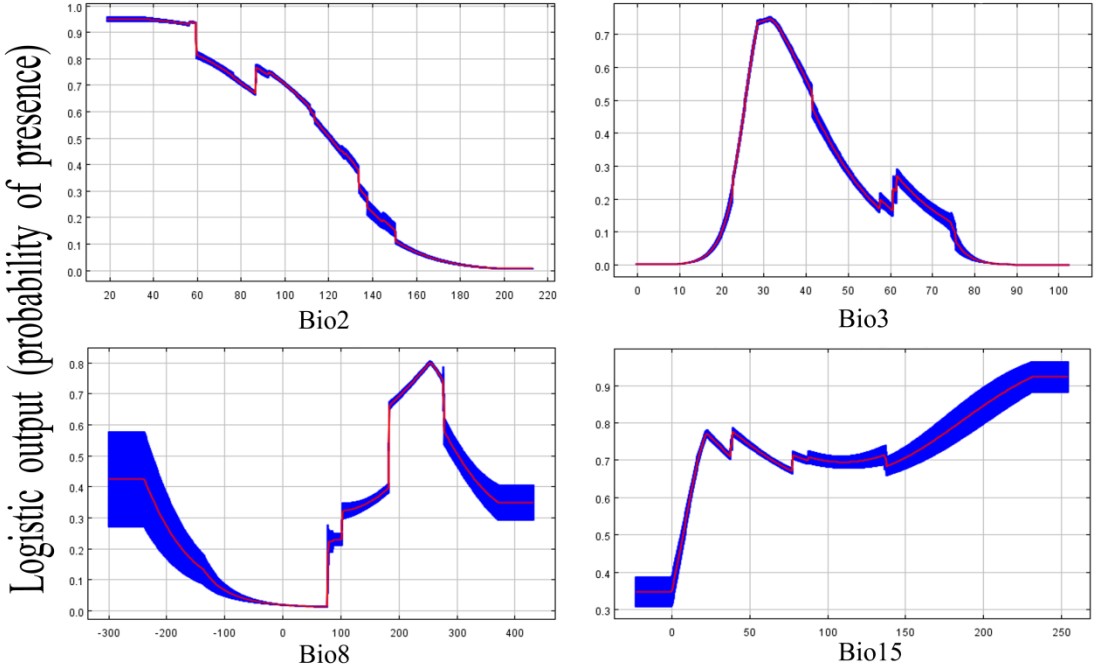

**Figure 3.** Response curves showing the relationships between the probability of presence of *P. pentagona* and four bioclimatic variables. Values shown are average over 10 replicate runs: blue margins show ± SD calculated over 10 replicates.

### 3.3. Future Climate Predictions

We next predicted potential distribution of *P. pentagona* under the Climate Change Scenarios and $CO_2$ emission scenarios (RCP2.6, RCP4.5, RCP6.0, RCP8.5) for 2050 (Figure 4) and 2070 (Figure 5) in Table S6. The model predicted an increase in the area of suitable habitat for 2050 and 2070 under RCP2.6, RCP4.5, RCP6.0, RCP8.5 compared to the current conditions. The suitable area for *P. pentagona* in 2050 using emission scenarios RCP2.6, RCP4.5, RCP6.0, RCP8.5 respectively were around $-2.55 \times 10^7$ km², $-2.60 \times 10^7$ km², $-2.62 \times 10^7$ km², and $-2.60 \times 10^7$ km², which predicted a 12.6%, 15.0%, 15.6%, and 15.0% increase in distribution compared to current conditions. For 2070, RCP2.6, RCP4.5, RCP6.0, RCP8.5 scenarios predicted $-2.60 \times 10^7$ km², $-2.76 \times 10^7$ km², $-2.70 \times 10^7$ km², and $-2.82 \times 10^7$ km², leading to a 14.6%, 21.7%, 19.1%, and 24.5% increase in suitable habitat compared to current conditions. The models demonstrate that increased temperature and carbon dioxide concentration will expand the area of suitable for *P. pentagona* around the world.

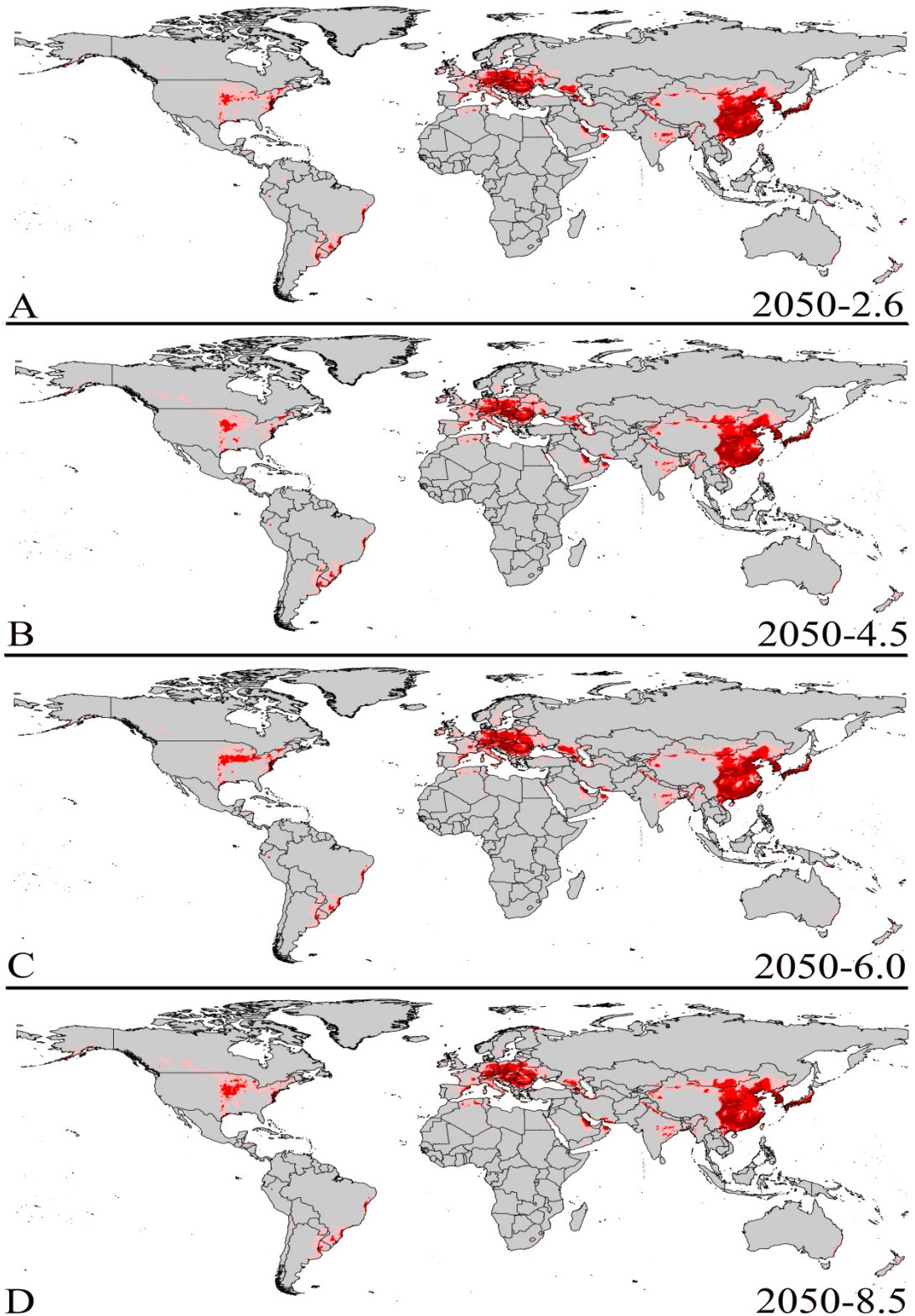

**Figure 4.** Future distribution models of *Pseudaulacaspis pentagona* on globe under different climate scenarios predicated by MaxEnt. Gray, unsuitable habitat area; Pale red, low habitat suitability area; Red, moderate habitat suitability area; Dark red, highly habitat suitability area. RCP: Representative Concentration Pathway. (**A**) RCP 2050-2.6; (**B**) RCP 2050-4.5; (**C**) RCP 2050-6.0; (**D**) RCP 2050-8.5.

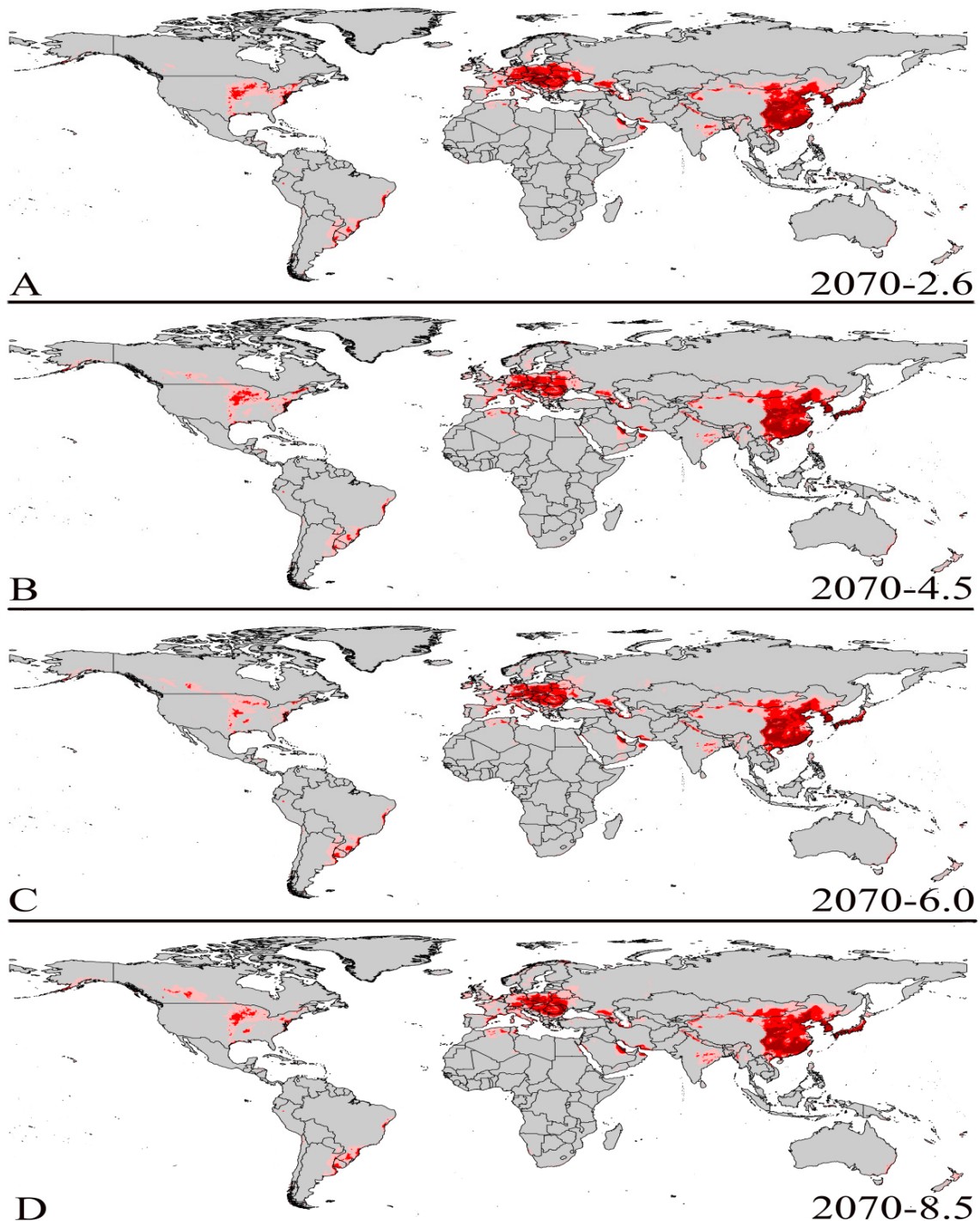

**Figure 5.** Future distribution models of *Pseudaulacaspis pentagona* on globe under different climate scenarios predicated by MaxEnt. Gray, unsuitable habitat area; Pale red, low habitat suitability area; Red, moderate habitat suitability area; Dark red, highly habitat suitability area. RCP: Representative Concentration Pathway. (**A**) RCP 2070-2.6; (**B**) RCP 2070-4.5; (**C**) RCP 2070-6.0; (**D**) RCP 2070-8.5.

## 3.4. Four Regions Future Climate Predictions

### 3.4.1. North America

In North America (Figure 6), the model indicated that the high habitat suitability area for 2050 of *P. pentagona* under the emission scenarios RCP2.6, RCP4.5, RCP6.0, RCP8.5 were $-4.50 \times 10^4$ km$^2$, $-2.83 \times 10^4$ km$^2$, $-5.14 \times 10^4$ km$^2$, and $-6.03 \times 10^4$ km$^2$, which was 75.2%, 84.4%, 71.7%, and 66.7% lower than the current conditions. For 2070, the emission scenarios RCP2.6, RCP4.5, RCP6.0, RCP8.5 predicted habitat areas of $-7.82 \times 10^4$ km$^2$, $-3.89 \times 10^4$ km$^2$, $-2.79 \times 10^4$ km$^2$, and $-2.60 \times 10^4$ km$^2$, which was 56.9%, 78.6%, 84.6%, and 85.6% lower than current conditions. The model predicted that change climate, regions with high habitat suitability for *P. pentagona* would decrease in North America. However, the predicted area gains in suitable habitat were predicted to be about $-4.48 \times 106$km$^2$, which was 13.9% higher than the current conditions when compared to RCP 8.5-2050, and compared to RCP 8.5-2070, where the suitable habitat was predicted to be about $-4.90 \times 10^6$ km$^2$ (24.8% increase). These results indicate that climate change would increase the risk of damages caused by *P. pentagona* in North America (Figure 2B).

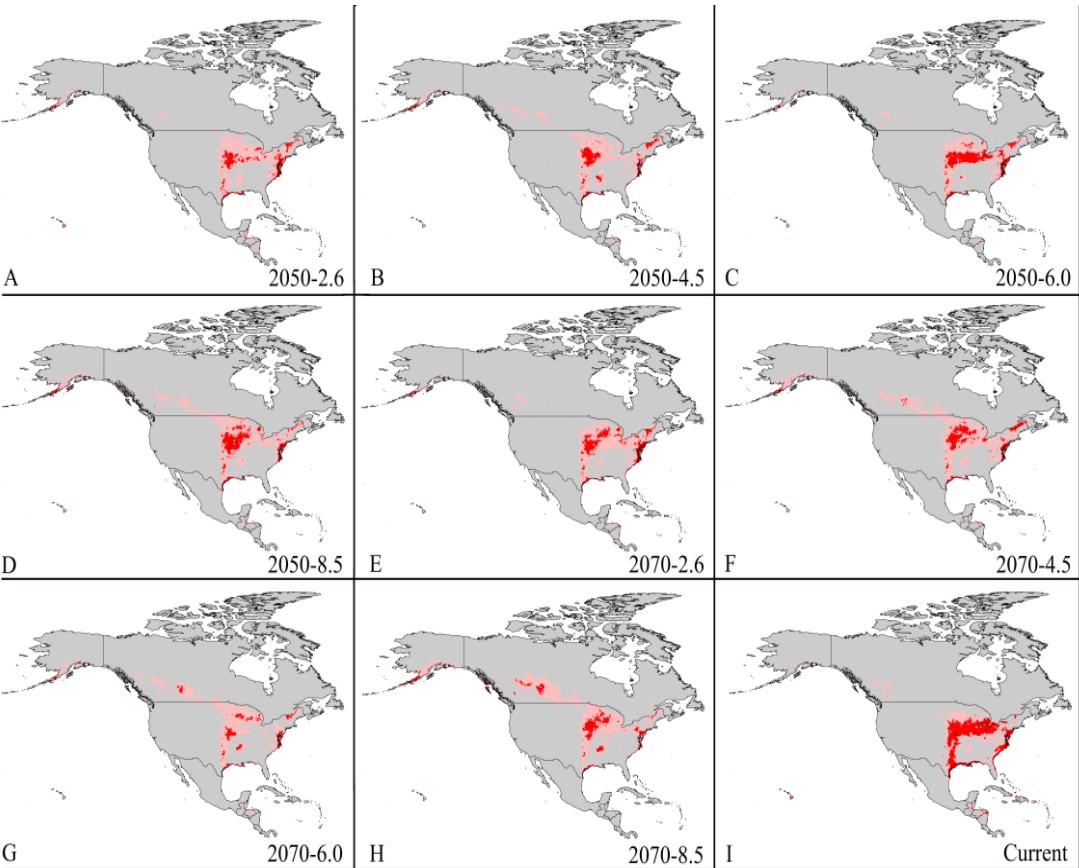

**Figure 6.** Future potential suitable habitats of *Pseudaulacaspis pentagona* on the North America continent under different climate scenarios predicated by MaxEnt. Gray, unsuitable habitat area; Pale red, low habitat suitability area; Red, moderate habitat suitability area; Dark red, high habitat suitability area. (**A**) RCP 2050-2.6; (**B**) RCP 2050-4.5; (**C**) RCP 2050-6.0; (**D**) RCP 2050-8.5; (**E**) RCP 2070-2.6; (**F**) RCP 2070-4.5; (**G**) RCP 2070-6.0; (**H**) RCP 2070-8.5; (**I**) Current.

### 3.4.2. Europe

For Europe (Figure 7), the predicted area gains in suitable habitat and high habitat suitability both increased under future climate change scenarios. Under RCP 6.0-2050, MaxEnt predicted that the suitable area to be about −6.48 × 10$^6$ km$^2$, approximately 29.9% higher than today. Under RCP 4.5-2070, the suitable area was predicted to be −6.47×10$^6$ km$^2$, a 29.8% increase. Under RCP8.5-2050, the high habitat suitability area was 17.1% larger than current conditions, reaching about −6.92 × 10$^5$ km$^2$. For RCP2.6-2070, the high habitat suitability region was predicted to be −8.64 × 10$^5$ km$^2$, 46.2% higher than current conditions (Figure 2C). In Europe, high, moderate and marginally suitable areas for *P. pentagona* were predicted to continuously increase until 2070.

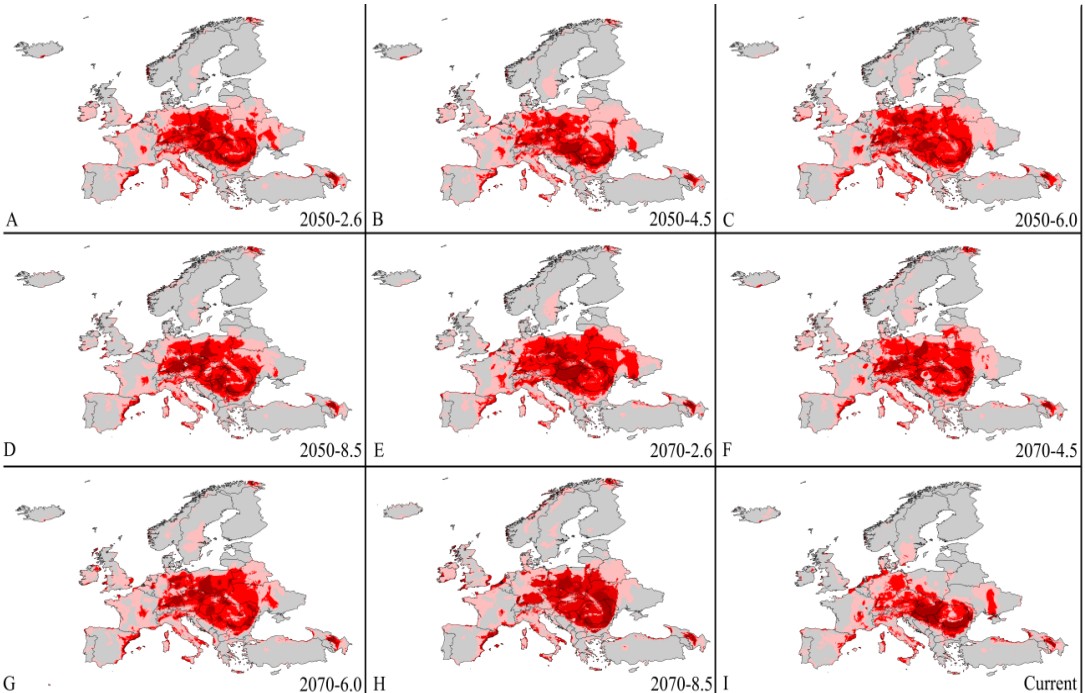

**Figure 7.** Future potential suitable habitats of *Pseudaulacaspis pentagona* on the Europe continent under different climate scenarios predicated by MaxEnt. Gray, unsuitable habitat area; Pale red, low habitat suitability area; Red, moderate habitat suitability area; Dark red, high habitat suitability area. (**A**) RCP 2050-2.6; (**B**) RCP 2050-4.5; (**C**) RCP 2050-6.0; (**D**) RCP 2050-8.5; (**E**) RCP 2070-2.6; (**F**) RCP 2070-4.5; (**G**) RCP 2070-6.0; (**H**) RCP 2070-8.5; (**I**) Current.

### 3.4.3. China and Korean Peninsula

In China (Figure 8), almost all of the Southeastern, South-central, Southwestern and Central regions and Huabei regions are currently highly suitable for *P. pentagona*. MaxEnt models showed that the total area of suitable land will slightly increase in the future, although the suitable area will shift from south to north towards Jilin and Heilongjiang Provinces. Suitable land will include scattered distributions in many provinces, including Xinjiang province. Under RCP8.5-2070, suitable area reached the largest, −7.89 × 10$^6$ km$^2$, which is about 24.1% higher than what is observed today (Figure 2D).

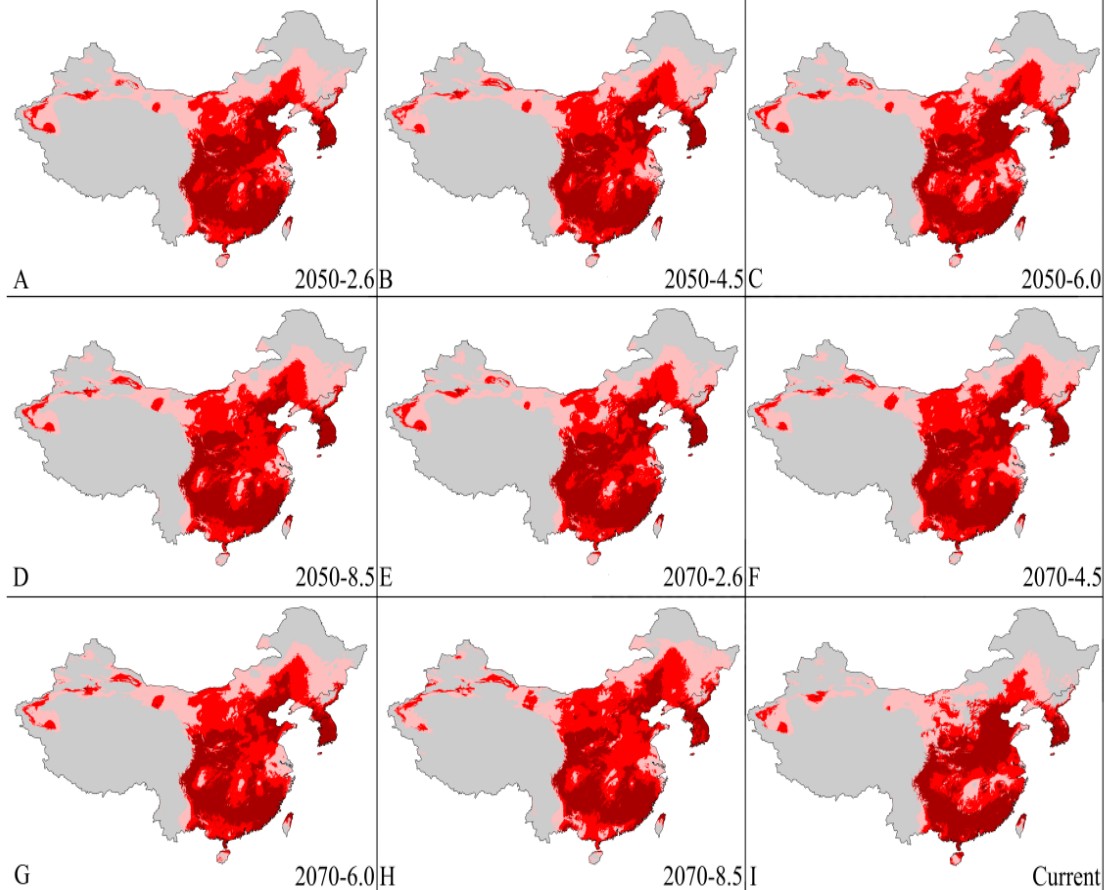

**Figure 8.** Future potential suitable habitats of *Pseudaulacaspis pentagona* on China and Korean Peninsula under different climate scenarios predicated by MaxEnt. Gray, unsuitable habitat area; Pale red, low habitat suitability area; Red, moderate habitat suitability area; Dark red, high habitat suitability area. (**A**) RCP 2050-2.6; (**B**) RCP 2050-4.5; (**C**) RCP 2050-6.0; (**D**) RCP 2050-8.5; (**E**) RCP 2070-2.6; (**F**) RCP 2070-4.5; (**G**) RCP 2070-6.0; (**H**) RCP 2070-8.5; (**I**) Current.

### 3.4.4. Japan

In Japan (Figure 9), the model predicts that the proportion and distribution of suitable areas and highly suitable areas will continuously increase from 2050–2070 regardless of the specific climate model. RCP6.0-2050 and RCP8.5-2070 predicted the largest increase in habitat for *P. pentagona*, increasing by 23.8% ($-2.66 \times 10^5$ km$^2$) and 20.3% ($-4.60 \times 10^5$ km$^2$), respectively (Figure 2E).

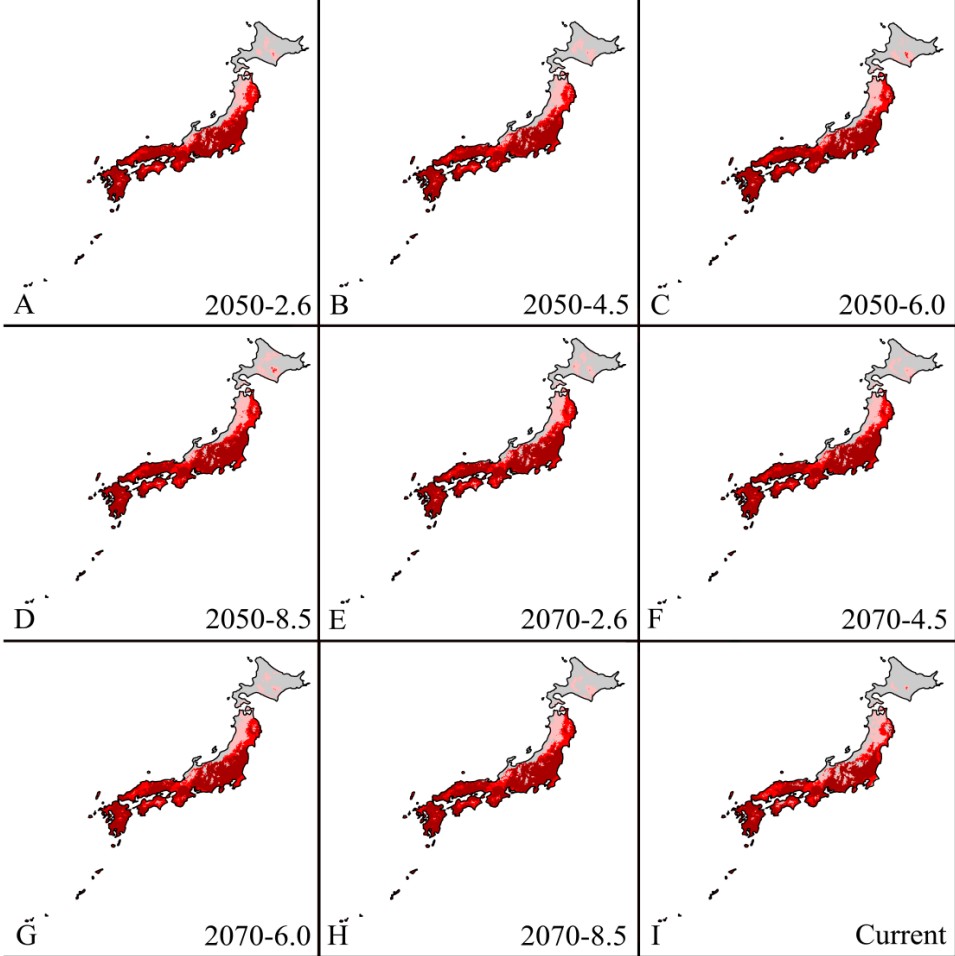

**Figure 9.** Future potential suitable habitats of *Pseudaulacaspis pentagona* on Japan continent under different climate scenarios predicated by MaxEnt. Pale red, unsuitable habitat area; Red, low habitat suitability area; Dark red, moderate habitat suitability area; Red, high habitat suitability area. (**A**) RCP 2050-2.6; (**B**) RCP 2050-4.5; (**C**) RCP 2050-6.0; (**D**) RCP 2050-8.5; (**E**) RCP 2070-2.6; (**F**) RCP 2070-4.5; (**G**) RCP 2070-6.0; (**H**) RCP 2070-8.5; (**I**) Current.

*3.5. Performance for the Predictive Model and Key Factors for Creating a Suitable Place*

We simulated the global occurrences of *P. pentagona* using MaxEnt to predict the potential distribution patterns of *P. pentagona* under current and future climate scenarios. Our models predict that climate change would affect the worldwide distribution of *P. pentagona*. Model also found that *P. pentagona* has a high invasive potential, particularly in the Central North America, Eastern South America, Central Europe, Eastern China and Japan. The areas unsuitable for *P. pentagona* in the current distribution map coincides with cold and dry areas, such as colder parts of Russia and Canada, dry areas of Australia and Egypt. The model found that the most important factor that influenced the current distribution of *P. pentagona* was temperature. For predicting future distribution, Bio2 (Mean diurnal range) and Bio3 (isothermality) contributed 77.4% percent towards the modeling of the current distribution of this species (Table 1).

**Table 1.** Relative contribution of each environmental variables to MaxEnt model.

| Environment Variables | Relative Contribution |
|:---:|:---:|
| BIO3 | 48.6% |
| BIO8 | 28.8% |
| BIO2 | 20.1% |
| BIO15 | 2.5% |

The MaxEnt model's predictions for current areas that are highly suitable for *P. pentagona* generally correlated with known records of *P. pentagona* in the literature. Prediction models for future distribution in the 2050s and 2070s were obtained using different climate change scenarios put forward by IPCC. These models indicated that climatic change would significantly affect the distribution of suitable habitat for *P. pentagona*. In general, we found that global warming would likely lead to the expansion of highly and moderately suitable habitats for *P. pentagona* in high latitude areas in the North. In contrast, suitable regions were reduced in some countries. Taken together, suitable regions worldwide would increase under current models of climate change.

## 4. Discussion

The current map of *P. pentagona* predicted potential distribution suggests that it has a wider distribution range than other species known potential distribution areas, such as *P. solenopsis* [34] and *P. madeirensis* Green [45]. Due to one of the most complex feeding habitats in *Diaspididae* [56], it will have a significantly larger distribution range in the future compared to other scale insects, and more harmful. *P. pentagona* is considered the primary pest for peach trees in the East Mediterranean region and in the Black Sea region. It is one of the most widely distributed insect species in orchards in Turkey [13]. *P. pentagona* also has a very high incidence in Kiwifruit orchards in Northern Iran [8]. According to relevant literature records, *P. pentagona* often occurs in peach and mulberry trees in China. The continued harm of *P. pentagona* will seriously affect the quality of agricultural production in China and the European countries along the Mediterranean coast, and cause serious economic losses. This is also consistent with our potential distribution of predictions. Although there are occurrence records in SE USA and Central and South America, they are not necessarily highly habitat suitability area. Scattered occurrence records may be carried by human transport and these places may not be able to survive. There are many sources of occurrence data collected by several people, so the reliability of some data is not particularly high, and there are some errors.

Potential distribution maps suggest that the most suitable habitats in Japan, China, and Europe are connected with the current location of *P. pentagona*. The most suitable habitats in North America and the east coast of South America are isolated. Potential distribution maps suggest that the most suitable habitats in Japan, China, and Europe are connected with the current location of *P. pentagona*. The most suitable habitats in North America and the east coast of South America are isolated. The model predicted that there would be a reduction of high and suitable ranges in America under future climate scenarios compared to current conditions. However, in other continents such as Asia and Europe, there would be a significantly expansion of high and suitable ranges under future climate scenarios RCP 8.5-2050, and RCP 8.5-2070. In result, this would lead to a significant expansion of the suitable area of *P. pentagona* in the global habitat.

The shift in suitable regions offers important insights for scientists, farmers and governments to prevent the expansion of *P. pentagona* and manage its occurrence. Previous studies on this species have focused on classification [54], molecular identification, physiological characteristics [14], and the chemical and biological control of *P. pentagona* [8,10,11]. Very few studies have investigated the effects of climatic change on the distribution of *P. pentagona*. We used models to predict regions that will have high risk of invasion by *P. pentagona*. Our model provides preliminary evidence for developing surveillance strategies to detect future infections by government agencies in currently uninfected areas. Finally, we formulated corresponding measures based on different levels of *P. pentagona* in the risk map [55].

For example, our analysis suggests that the Black Sea coast of southern Ukraine, the junction of Russia and Georgia, Central Georgia and Azerbaijan will be highly suitable environments for *P. pentagona* under climate change. In regions that are predicted to become highly suitable habitats for *P. pentagona*, government agencies should develop strict quarantine measures to prevent expansion by human activities, such as ornamental plant and agriculture product importation [57]. In areas of moderate habitat suitability, government agencies should strengthen the monitoring of insect population density to prevent the occurrence of *P. pentagona* and prevent a large-scale outbreak. Governments can advise farmers to plant crops that are not susceptible to *P. pentagona* in highly suitable region to increase yield and reduce losses [56,58]. The study not only provides references for further establishing the distribution range of *P. pentagona*, but also provides a theoretical basis for determining management and quarantine strategies for this pest.

Our model has some disadvantages. Firstly, because the occurrence data was collected from published work, we were unable to identify the actual environmental conditions for each collection point, resulting in small deviations of the simulation results. Secondly, our model only considers the impact of climate change on the change in the distribution area, without considering the relationship of *P. pentagona* and other species. Our model does not take into account the effects of altitude, host distribution, inter-species interactions, and human activity index. The above factors will likely have some effect on species distribution. We would continue to consider these factors in future research in the hope of improving the accuracy of model prediction. However, the model yielded optimal results for partial ROC (mean value AUC: 0.9526019), indicating that it has a strong fit based on the current distribution of this species. Moreover, accessible area and unsuitable region are play a crucial role in ecological niche modeling and species distribution modeling, especially for a huge extent, for example, globally [59,60]. These factors will improve the realism and accuracy of distribution projections when combining the accessible area and unsuitable regions in create model. Thus, these factors would be considered in future study.

In summary, our study provides important information on the potential distribution of *P. pentagona* under climate change and provides important insights into the management of this species.

## 5. Conclusions

Climate change clearly affects the distribution pattern of invasive species. The current study investigated and predicted the potential global distribution of *P. pentagona* under current and future climate change scenarios. The results demonstrated the areas of climatic suitability would be larger than current condition, especially in Eastern Asia and Europe. The present study provides a reference to the pest develop policies for its control.

**Supplementary Materials:** The following are available online at http://www.mdpi.com/1999-4907/11/2/192/s1, Figure S1: Performances of niche model of pest., Figure S2: Partial AUC Values and Graphics, null model (red distribution), distribution of expectations created via bootstrapping replacement of 50% of the total available points and 1000 resampling replicates (blue distritbuion), Table S1: Distribution sites for *P. pentagona*. Table S2: References used to compile the dataset, Table S3: After filtering distribution sites for *P. pentagona*, Table S4: Pearosn correlation coefficients matrix of *P. pentagona*, Table S5: ENMeval results for *P.pentagona* from SDMs, Table S6: Future distribution models of *Pseudaulacaspis pentagona* on globe under different climate scenarios predicated by MaxEnt on 2050s and 2070s.

**Author Contributions:** Conceptualization, Y.L. and L.Z.; methodology, Y.L.; software, Y.L. and J.W.; investigation, Y.L.; resources, Y.L.; data curation, Y.L. and Q.Z.; writing—original draft preparation, Y.L. and J.W.; writing—review and editing, Y.L. and L.C.; project administration, J.W.; funding acquisition, J.W., Q.Z. and H.Z. All authors have read and agreed to the published version of the manuscript.

**Funding:** This project was supported by the National Natural Science Foundation of China (NSFC) grant (No. 31301899, No. 31501876 and No. 31872272) and Shanxi Agricultural University of Science and Technology Innovation fund projects (2015YJ03). The funders had no role in study design, data collection and analysis, decision to publish, or preparation of the manuscript.

**Acknowledgments:** We thank local plant protection stations of agricultural department for their valuable fieldwork and collection of monitoring data. We thank anonymous reviewers for their valuable comments.

**Conflicts of Interest:** The authors declare no conflict of interest. The funders had no role in the design of the study; in the collection, analyses, or interpretation of data; in the writing of the manuscript, or in the decision to publish the results.

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
