# Peer review of "The Potential Global Distribution of the White Peach Scale Pseudaulacaspis pentagona (Targioni Tozzetti) under Climate Change"

_forests, doi:10.3390/f11020192_

Round 1

Reviewer 1 Report

In this manuscript, Lu et al develop a MaxEnt species distribution model (SDM) for white peach scale – a major economic pest.

Overall the manuscript is well written (though the Discussion is too short and lacking much reference to the wider literature), but could use proof reading by a native English speaker in places where some unusual grammar is used.

The scope of the study is very much in terms of using standard SDMs to model a specific pest species and make climate change projections, rather than develop new conceptual or methodological approaches. Overall, the modelling follows standard practice. However I do have some technical criticisms, which the authors may consider investigating.

I was pleased to see that the modellers accounted for recording biases using both a filtering or thinning of their records and a weighted target group approach. However, when reading section 2.1 it was not clear initially that both approaches were combined. I suggest re-phrasing section 2.1 (line 101) to make clear that the filtering was done to thin over-sampled localities. Having said that, unless filtering was applied to each individual species in the target group data, was it actually necessary? In addition, more detail of the insect species used as the target group is needed (line 159).

My strongest criticism concerns the selection of climatic predictors based on low correlation between the Bioclimatic variables. In my view, this leads to a rather strange selection of predictors that are unlikely to represent the major limiting factors on insect distributions. Instead of biologically intuitive variables like temp of warmest quarter (correlated to degree days for development) or min temp of coldest month (winter survival) the variables used in the modelling are mean diurnal temperature range, isothermality, temp of wettest quarter and precipitation seasonality. The authors do not justify this choice in terms of insect biology and have not examined the sensitivity of their predictions to the choice of predictor variables. Unless this is done I do not feel the current or future projections are reliable. Indeed, I am surprised by some features of the maps, e.g. projection of unsuitability in SE USA and Central and South America.

Further, I would like to see partial response curves for the predictor variables in the model, so that readers can assess the realism of modelled responses from which projections are made.

Host plant land cover or other anthropogenic predictors are not considered in the modelling, but may have a strong influence on pest spread. This should be discussed as a limitation of the model in the Discussion.

The authors do not state the extent of the model, though I assume it was all land globally. As such, pseudo-absences/background points will be sampled globally rather than from the ‘accessible area’ of the species, usually defined as a buffer zone around the presence records. For invasive species this is critical as there are likely suitable regions where the species has not yet reached. If pseudo-absences are sampled there then the model will conflate absence through dispersal constraints to absence because of climatic unsuitability. This issue should at least be discussed and preferably amended in the modelling. See Barve et al (2011) The crucial role of the accessible area in ecological niche modeling and species distribution modelling. Ecological Modelling and Chapman et al (2019) Improving species distribution models for invasive non‐native species with biologically informed pseudo‐absence selection. Journal of Biogeography.

Overall, I feel the choice of predictor variables and use of a global modelling extent mean that the projected distributions are not sufficiently reliable to support strong conclusions or risk management for the pest.

Minor comments

L17 – change the distribution map to a potential distribution map

L21 - In the abstract it is necessary to define BIO3

L72 – typo for ManEnt

L73 – typo for envelope and also a missing space

There are far too many repetitive figures. I suggest presenting fewer future scenarios in the main text and moving much of the figures to Supporting Information.

Author Response

Dear Reviewers:

    Thank you for your letter and for the comments of reviewer concerning our manuscript entitled “The potential global distribution of the white peach scale Pseudaulacaspis pentagona (Targioni Tozzetti) under climate change” (ID: forests-691642). Those comments are all valuable and very useful for revising and improving our manuscript. Revised portion are marked in the paper and the main corrections in the paper. Some minor errors were correct in revised manuscript and a English language consultant was used in current manuscript. Addition, some question which reviewer’s suggestion were point to point responds are as flowing:

Responds to reviewer’s comments:

1. Comment 1: The Discussion is too short and lacking much reference to the wider literature.

Response to comment 1: Thanks for reviewer’s value comments. We have rewrite the discussion section and cited a wider reference.

2. Comment 2: The modellers accounted for recording biases using both a filtering or thinning of their records and a weighted target group approach. However, when reading section 2.1 it was not clear initially that both approaches were combined. I suggest re-phrasing section 2.1 (line 101) to make clear that the filtering was done to thin over-sampled localities. Having said that, unless filtering was applied to each individual species in the target group data, was it actually necessary? In addition, more detail of the insect species used as the target group is needed (line 159).

Response to comment 2: Thanks for value suggestion. We only used filtering to process data collection preferences. We were sorry about we did not adopt a weighted target group approach to process data collection. We have deleted this method in revised manuscript.

3. Comment 3: My strongest criticism concerns the selection of climatic predictors based on low correlation between the Bioclimatic variables. In my view, this leads to a rather strange selection of predictors that are unlikely to represent the major limiting factors on insect distributions. Instead of biologically intuitive variables like temp of warmest quarter (correlated to degree days for development) or min temp of coldest month (winter survival) the variables used in the modelling are mean diurnal temperature range, isothermality, temp of wettest quarter and precipitation seasonality. The authors do not justify this choice in terms of insect biology and have not examined the sensitivity of their predictions to the choice of predictor variables.

Response to comment 3: Thanks for value comments. Indeed, biologically intuitive variables might be represented the major limiting factors on insect distributions. However, excluded the highly correlated variables by PCA, Spearman or Pearson methods used in many taxa, such as plant (Xu et al., 2019; Zhang et al., 2019) and animal (Santana Jr et al., 2019) in small scale (country) (Xu et al., 2019)and large scale (global) (Santana jr et al., 2019; Cunze et al., 2019). So, we think this approach is appropriate in current study.

    Cunze S, Kochmann J, Slimpel S. 2019. Global occurrence data improve potential distribution models for Aedes japonicus japonicus in non-native regions. Pest management science (online).

    Xu DP, ZHuo ZH, Wang RL, Ye M, Pu B. 2019. Modeling the distribution of Zanthoxylum armatum in China with MaxEnt modeling. Global ecology and onservation. 19: e00691.

Zhang JJ, Jiang F, Li GY, Qin W, Li SQ, Gao HM, Cai ZY, Lin GH, Zhang TZ. 2019. Maxent modeling for predicting the spatial distribution of three raptors in the   Sanjiangyuan National Park, China. Ecology and Evolution, 9 (11): 6643-6654.

Santana Jr PA, Kumar L, Da Silva RS, Pereira JL, Picanco MC. 2019. Assessing the impact of climate change on the worldwide distribution of Dalbulus maidis (Delong) using MaxEnt. Pest management science, 75(10): 2706-2715.

4. Comment 4: I am surprised by some features of the maps, e.g. projection of unsuitability in SE USA and Central and South America.

Response to comment 4: Thanks for value comments. Although there are collection records in SE USA and Central and South America, they are not necessarily highly habitat suitability area. Scattered distribution records may be carried by human transport and may not be able to survive. There are many sources of distributed data collected by several people, so the reliability of some data is not particularly high, and there are some errors. We discuss this issue in the Discussion section.

5. Comment 5: Further, I would like to see partial response curves for the predictor variables in the model, so that readers can assess the realism of modelled responses from which projections are made.

Response to comment 5: Thanks for value comments. We have added partial response curves for the predictor variables in the model in revised manuscript.

6. Comment 6: Host plant land cover or other anthropogenic predictors are not considered in the modelling, but may have a strong influence on pest spread. This should be discussed as a limitation of the model in the Discussion.

Response to comment 6: Thanks for value suggestions. Our model does not take into account the effects of altitude, host distribution, inter-species interactions, and human activity index. We discuss this factors in the Discussion section in revised manuscript.

7. Comment 7: The authors do not state the extent of the model, though I assume it was all land globally. As such, pseudo-absences/background points will be sampled globally rather than from the ‘accessible area’ of the species, usually defined as a buffer zone around the presence records. For invasive species this is critical as there are likely suitable regions where the species has not yet reached. If pseudo-absences are sampled there then the model will conflate absence through dispersal constraints to absence because of climatic unsuitability. This issue should at least be discussed and preferably amended in the modelling. See Barve et al (2011) The crucial role of the accessible area in ecological niche modeling and species distribution modelling. Ecological Modelling and Chapman et al (2019) Improving species distribution models for invasive non‐native species with biologically informed pseudo‐absence selection. Journal of Biogeography.

Response to comment 7: Response to reviewer: Thanks for your value suggestion and careful work. Indeed, the extent of the model was all land globally. We discussed this issue in discuss part.

8. Comment 8 (Page 1, Line 17): Change the distribution map to a potential distribution map.

Response to comment 8: Thanks for reviewer’s value comments. We have changed the distribution map to a potential distribution map in revised manuscript.

9. Comment 9 (Page 1, Line 21): In the abstract it is necessary to define BIO3.

Response to comment 9: Thanks for your suggestions for this point. We have defined BIO3 in revised manuscript.

10. Comment 10 (Page 1, Line 72): Typo for ManEnt.

Response to comment 10: Thanks for your carefully work. We rewrite this word in revised manuscript.

11. Comment 11 (Page 1, Line 73): Typo for envelope and also a missing space.

Response to comment 11: Thanks for your carefully work. We rewrite this word and added a space in revised manuscript.

12. Comment 12There are far too many repetitive figures. I suggest presenting fewer future scenarios in the main text and moving much of the figures to Supporting Information.

Response to comment 12: Thanks for carefully work and suggestions. We have moved the future global distribution map of 2050 and 2070 to supporting materials Table S6.

Reviewer 2 Report

Authors analyzed current distribution of white peach scale and predicted its future changes in distribution, according to projected climate changes. Due to economic importance of this invasive pathogen, study aim and results are important for both science and practice. I am not sure whether the study fits to the aims and scope of ‘Forests’ – Authors did not refer to forest ecosystems, but rather to orchards and plantations. However, in my opinion study is important for tree biology and ecology. Authors carefully performed analyses and documented all steps in a way allowing for replication. Results are extensively described and needs shortening. Discussion requires improvements, especially in terms of results implications and its significance.

Specific comments:

Abstract – in my opinion too much space in abstract is devoted to rationale study aims, and too small – to results. Please remember that many people reads only abstract or use it to cite a paper. I’d add more information about results and conclusions.

12 – please add Hemiptera: Diaspididae after name – it will be more descriptive for people not familiar with species name

16 – term ‘endemic’ suggests that species occurs only in one region, here its an oxymoron – it could be endemic to Japan if not occur in China, it would be better to write ‘native to Japan and China’

19-20 – too general – maybe better specify particular regions or how much percent of such continent is within a potential range? How much it will increase in optimistic and pessimistic scenario?

Keywords – please avoid repeating keywords from title. Perhaps phrases like ‘risk assessment’, ‘species distribution model’, ‘habitat suitability’, ‘pest management’ would increase the findability and citability of the study.

Introduction

30 – ‘non-endemic’ may also be native, invasive is assumed to be non-native (alien) species. Change into ‘non-native’, ‘exotic’ or ‘alien’

36 – please delete the space before comma

42 – not clear what are ‘landscape plants’ and ‘wild plants’ – please reword or specify. Is the species studied a potential threat to forest trees? Here it would be worthy to state how this species influence forests.

44 – please check grammar in this sentence – unclear

56-57 – repetition of information included earlier – please omit

69-78 -  rationale for method used (MaxEnt) fits better to methods section

Methods

97 – ‘accurate to county’ – not clear – did Authors mean that the resolution was of county level?

106 – please divide ESRI and 2012

124 – please divide Table and S4. Also, in file with Table S4 caption is P. madeirensis, instead of P. pentagona. I’d also change caption to Pearosn correlation coefficients matrix of...

125-135 – Authors used three GCMs to reduce uncertainity in models, however it is not clear how Authors accounted for all of them. Did Authors averaged climatic layers, or output of models applied to each GCM? I know that this is in lines 175-177, but such questions raised when reading 125-135. Maybe move these lines here?

140 – please explain abbreviation ENM

Results

in various places – area is expressed as xxx106 or 107 km2 and 6 and 7 are not superscript – please carefully read and correct this.

Fig. 1 – using differential colors suggests lack of order between them – maybe continuous scale (e.g. grey, pale red, red, dark red) would be more informative and intuitive?

3.3. – it would be better to join this section with 3.4. Further part of results is very descriptive – please try to make it more concise

245-246 – Japan is also East Asia – please reword, maybe China and Korean Peninsula would be better (looking on maps)?

Fig. 9 and 10 – China and Korea are not a continent – please correct the caption

Discussion

Overall, discussion is the weakest part of the manuscript. Authors should focus more on comparison of predicted spread with results obtained for other pest species, discuss more consequences and potential actions to take in particular countries. Some questions which might be interesting:

- Is the potential spread of species studied higher or lower to other pests with known future distribution? Is this more or less danger then these species?

- How future distribution is connected with places with infested species orchards? How it might impact economy of particular countries?

- Which factor are crucial for species studied spread?

- How to  prevent invasion of this species?

- Limitation – dispersal – how it might influence results? How much of further spread is climate-related, and how much needs human activity (transport)? How it influences real future distributions (which areas predicted to be suitable are isolated, and which are connected with current species distribution)?

- application – how species current and future distribution assessment allow for better management and prevention of invasive pests? How it may improve management?

334-354 – these two paragraphs are result, not discussion. This describes the second aim of the study – factors determining the occurrence of species studied.

Conclusions

384 – add ‘i’ to ‘nvestigated’

Author Response

Dear Reviewer:

Thank you for your letter and for the comments of reviewer concerning our manuscript entitled “The potential global distribution of the white peach scale Pseudaulacaspis pentagona (Targioni Tozzetti) under climate change” (ID: forests-691642). Those comments are all valuable and very useful for revising and improving our manuscript. Revised portion are marked in the paper and the main corrections in the paper. Some minor errors were correct in revised manuscript and a English language consultant was used in current manuscript. Addition, some question which reviewer’s suggestion were point to point responds are as flowing:

Responds to reviewer’s comments:

1. Comment 1: Abstract – in my opinion too much space in abstract is devoted to rationale study aims, and too small – to results. Please remember that many people reads only abstract or use it to cite a paper. I’d add more information about results and conclusions.

Response to comment 1: Thanks for your value comment. We've added more about the research results in the abstract so that more people can understand the content and use it to cite the paper.

2. Comment 2 (Page 1, Line 12): Please add Hemiptera: Diaspididae after name – it will be more descriptive for people not familiar with species name.

Response to comment 2: Thanks for your comment. We've added“Hemiptera: Diaspididae” after species name in revised manuscript.

3. Comment 3 (Page 1, Line 16): Term ‘endemic’ suggests that species occurs only in one region, here its an oxymoron – it could be endemic to Japan if not occur in China, it would be better to write ‘native to Japan and China’.

Response to comment 3: Thanks for your value suggestion. Due to this pest appears in both China and Japan, we have corrected ‘endic’ to ‘native’.

4. Comment 4 (Page 1, Line 19-20): Too general – maybe better specify particular regions or how much percent of such continent is within a potential range? How much it will increase in optimistic and pessimistic scenario?

Response to comment 4: Thanks for your suggestion. We have described our experimental results in detail in revised manuscript.

5. Comment 5: Keywords – please avoid repeating keywords from title.

Response to comment 5: Thanks for your value suggestion. In order to increase the findability and citability of the study, we have replaced the ‘climate change’, ‘MaxEnt’, ‘global’ in keywords with ‘species distribution model’, ‘risk assessment’, ‘habitat suitability’, ‘pest management’.

6. Comment 6 (Page 1, Line 30): ‘Non-endemic’ may also be native, invasive is assumed to be non-native (alien) species. Change into ‘non-native’, ‘exotic’ or ‘alien’.

Response to comment 6: Thanks for your value comment. Yes, you are right. We are very sorry for our ambiguous expression. We have changed the ‘Non-endemic’ to ‘non-native’.

7. Comment 7 (Page 1, Line 36): Please delete the space before comma.

Response to comment 7: We feel great thanks for your professional review work on our article. We have deleted the space before comma.

8. Comment 8 (Page 1, Line 42): Not clear what are ‘landscape plants’ and ‘wild plants’ – please reword or specify. Is the species studied a potential threat to forest trees? Here it would be worthy to state how this species influence forests.

Response to comment 8: Thanks for your comment. We have rewrite and given examples, and explained how this species influence forests.

9. Comment 9 (Page 2, Line 44): Please check grammar in this sentence – unclear.

Response to comment 9: We are very sorry for our expression. We modified our sentence in revised manuscript.

10. Comment 10 (Page 2, Line 56-57): Repetition of information included earlier – please omit.

Response to comment 10: Thanks for your carefully work. We have omitted the corresponding duplicate information.

11. Comment 11 (Page 2, Line 69-78): Rationale for method used (MaxEnt) fits better to methods section.

Response to comment 11: Thanks for your value suggestion. We have added the rationale part about method used (MaxEnt) in page4, line171.

12. Comment 12 (Page 3, Line 97): ‘Accurate to county’ – not clear – did Authors mean that the resolution was of county level?

Response to comment 12: We are very sorry for our unclear expression. What we mean is that the accuracy of where the collection occur is to the county level.

13. Comment 13 (Page 3, Line 106): Please divide ESRI and 2012.

Response to comment 13: Thanks for your carefully work. We have divided ESRI and 2012.

14. Comment 14 (Page 3, Line 106): Please divide Table and S4. Also, in file with Table S4 caption is madeirensis, instead of P. pentagona. We’d also change caption to Pearosn correlation coefficients matrix of...

Response to comment 14: Thanks for your carefully work. We have completed the correction in revised manuscript.

15. Comment 15 (Page 3, Line 125-135): Authors used three GCMs to reduce uncertainity in models, however it is not clear how Authors accounted for all of them. Did Authors averaged climatic layers, or output of models applied to each GCM? I know that this is in lines 175-177, but such questions raised when reading 125-135. Maybe move these lines here?

Response to comment 15: Thanks for your comment. We have explained the reasons we used three GCMs in revised manuscript. We have adjusted as requested.

16. Comment 16 (Page 3, Line 140): Please explain abbreviation ENM.

Response to comment 16: Thanks for your value comment. We have explained abbreviation ENM in revised manuscript.

17. Comment 17: In various places – area is expressed as xxx106 or 107 km2 and 6 and 7 are not superscript – please carefully read and correct this.

Response to comment 17: Thanks for your carefully work. We have read and corrected carefully, and superscripted the numbers.

18. Comment 18 (Page 3, Line 140): Using differential colors suggests lack of order between them – maybe continuous scale (e.g. grey, pale red, red, dark red) would be more informative and intuitive?

Response to comment 18: Thanks for your comment. We have redone all the potential distribution maps and reconstructed the pictures with continuous scale.

19. Comment 19:3. – it would be better to join this section with 3.4. Further part of results is very descriptive – please try to make it more concise.

Response to comment 19: Thanks for your value comment. We have join this section with 3.4 after 3.3. We have described the results more concisely.

20. Comment 20 (Page 3, Line 140): Japan is also East Asia – please reword, maybe China and Korean Peninsula would be better (looking on maps)?

Response to comment 19: Thanks for your value comment. We modified the statement about four regions in revised manuscript.

21. Comment 21: 9 and 10 – China and Korea are not a continent – please correct the caption.

Response to comment 20: Thanks for your comment. We have corrected the caption about China and Korea regions.

22. Comment 22: Is the potential spread of species studied higher or lower to other pests with known future distribution? Is this more or less danger then these species?

Response to comment 22: Thanks for your value comment. Due to the most complex feeding habitat, it will have a significantly larger distribution range in the future compared to other scale insects, and more harmful.

23. Comment 23: How future distribution is connected with places with infested species orchards? How it might impact economy of particular countries?

Response to comment 23: Thanks for your comment. P. pentagona is considered the primary pest for peach trees in the East Mediterranean region and in the Black Sea region. It is one of the most widely distributed insect species in orchards in Turkey. P. pentagona also has a very high incidence in Kiwifruit orchards in Northern Iran. According to relevant literature records, P. pentagona often occurs in peach and mulberry trees in China. The continued harm of P. pentagona will seriously affect the quality of agricultural production in China and the European countries along the Mediterranean coast, and cause serious economic losses.

24. Comment 24: Which factor are crucial for species studied spread?

Response to comment 24: Previous research that has shown that the population growth of insect was mainly affected by temperature. This is also consistent with our model, three of the selected environmental variables are temperature, and BIO3 (Isothermality) contributes the most.

25. Comment 25: How to prevent invasion of this species?

Response to comment 25: Thanks for value comment. we suggest that special quarantine measure should be taken to limit potential expansion for high and suitable region for future infestations by human activities such as ornamental plant and agriculture product importation. Our model provides preliminary evidence for developing surveillance strategies to detect future infections by government agencies in currently uninfected areas.

26. Comment 26: Limitation – dispersal – how it might influence results? How much of further spread is climate-related, and how much needs human activity (transport)? How it influences real future distributions (which areas predicted to be suitable are isolated, and which are connected with current species distribution)?

Response to comment 26: Limitation: Our model does not take into account the effects of altitude, host distribution, inter-species interactions, and human activity index. These factors will affect the prediction of potential distribution areas. We will continue to consider these factors in future research in the hope of improving the accuracy of model predictions. The future growth of potential distribution areas is all climate-related. Potential distribution maps suggests most suitable habitats in Japan, China, and Europe are connected with the current location of P.pentagona. Most suitable habitats in North America and east coast of South America are isolated.

27. Comment 27: Application – how species current and future distribution assessment allow for better management and prevention of invasive pests? How it may improve management?

Response to comment 27: In areas of moderate habitat suitability, government agencies should strengthen the monitoring of insect population density to prevent the occurrence of P. pentagona and prevent a large-scale outbreak. Governments can advise farmers to plant crops that are not susceptible to P. pentagona in highly suitable region to increase yield and reduce losses

28. Comment 28 (Page 15-16, Line 334-354): These two paragraphs are result, not discussion. This describes the second aim of the study – factors determining the occurrence of species studied.

Response to comment 28: Thanks for your useful comment. We have adjusted these two paragraphs to the result section in the revised manuscript.

29. Comment 29 (Page 16, Line 384): Add ‘i’ to ‘nvestigated’.

Response to comment 29: Thanks for your careful work. We have rewrite the word in the revised manuscript.

Round 2

Reviewer 2 Report

Authors carefully revised the second version of the manuscript, detailly addressing all issues raised in my previous report. They also very clearly reported all changes, making second round of revision easier. I am thankful for that. I consider manuscript is OK now.

Specific comments:

72 – Zamia, Ilmus and Spondias are genera names – please add spp. to each of them

Author Response

Dear Reviewer:

    Thank you for your letter and for the comments of reviewer concerning our manuscript entitled “The potential global distribution of the white peach scale Pseudaulacaspis pentagona (Targioni Tozzetti) under climate change” (ID: forests-691642). Those comments are all valuable and very useful for revising and improving our manuscript. Revised portion are marked in the paper and the main corrections in the paper. Some minor errors were correct in revised manuscript and a English language consultant was used in current manuscript. Addition, some question which reviewer’s suggestion were point to point responds are as flowing:

Responds to reviewer’s comments:

 1. Comment 1 (Page 2, Line 72): Zamia, Ulmus and Spondias are genera names – please add spp. to each of them.

Response to comment 1: Thanks for your value comment. We've added spp. after genera name Zamia, Ulmus and Spondias in revised manuscript.
